**www.cambridge.org/ext**

**Review**

Wildlife trade; poaching; overharvest; wildlife trafficking; sustainable use

**Corresponding author:**
Amy Hinsley;
Email: amy.hinsley@biology.ox.ac.uk

# Trading species to extinction: evidence of extinction linked to the wildlife trade

Amy Hinsley[1,2] 📷, Jasmin Willis[1], Abigail R. Dent[1], Rodrigo Oyanedel[3,4], Takahiro Kubo[1,2,5] and Daniel W. S. Challender[1,2]

[1]Department of Biology, University of Oxford, Oxford, UK; [2]Oxford Martin Programme on Wildlife Trade, Oxford Martin School, University of Oxford, Oxford, UK; [3]Instituto Milenio en Socio-Ecología Costera (SECOS), Santiago, Chile; [4]Centro de Investigación en Dinámica de Ecosistemas Marinos de Altas Latitudes (IDEAL), Valdivia, Chile and [5]Biodiversity Division, National Institute for Environmental Studies, Tsukuba, Japan

## Abstract

The link between unsustainable harvest of species for the wildlife trade and extinction is clear in some cases, but little is known about the number of species across taxonomic groups that have gone extinct because of trade-related factors, or future risks for traded species. We conducted a rapid review of published articles and species assessments on the IUCN Red List of Threatened Species with the aim of recording examples of extinctions that were attributed to trade. We found reports of extinctions linked, at least in part, to wildlife trade for 511 unique taxa. These include 294 reports of global extinctions, 25 extinctions in the wild, and 192 local extinctions. The majority of global/in the wild extinctions linked to trade (230) involved ray-finned fishes, primarily due to predation by introduced commercial species. Seventy-one of the 175 reported local extinctions of animal taxa linked to trade were mammals. Twenty-two global/in the wild extinctions and 16 local extinctions of plants were reportedly linked to trade. One fungal species was reported locally extinct due to over-harvesting for trade. Furthermore, 340 species were reported to be near-extinct linked to trade, 269 of which were animals, including several high-profile megafauna. Extinctions were linked to direct harvesting and/or indirect threats such as bycatch or invasive species introduced for trade, but often it was not possible to determine the relative role of trade-related threats in extinctions. Our results highlight the need for better data collection on trade-related extinction risk to understand its impacts and to inform more effective wildlife trade policy.

## Impact statement

Overexploitation – the harvest or extraction at a rate that exceeds the ability of populations to recover – is widely recognised as a major threat to biodiversity. Some overexploitation is principally for wildlife trade, as distinct from the use of wildlife at a subsistence level. The wildlife trade is frequently highlighted in contemporary conservation science literature and in the press as a key threat to species. While the extraction of species for the wildlife trade can be unsustainable in some cases, in others it can contribute to the conservation of species, for example, by providing economic incentives to conserve species and their habitats, and providing a range of benefits to people. This research contributes to a better and clearer understanding of the links between the wildlife trade and the extinction of species by elucidating how these links are characterised in the literature, including proximate threats, 'near-extinctions', and indirect drivers of extinction linked to the wildlife trade. The results are based on a review of the literature published in the period 1960 to 2021 and an examination of available information on Extinct species on the IUCN Red List of Threatened Species. The study concludes by providing recommendations on how the extinction of species linked to the harvest, use, and trade of wildlife could be improved in future research.

## Introduction

Biodiversity is under unprecedented threat from climate change, changing land and sea use, the overexploitation of natural resources, pollution, and invasive species (Maxwell et al., 2016; Tilman et al., 2017). Mitigating these threats requires understanding both their proximate and ultimate drivers, impacts on biodiversity and contribution to extinction risk, and how to intervene most effectively at different scales (Lenzen et al., 2012; 't Sas-Rolfes et al., 2019). Overexploitation has long been recognised as a threat to species (Broad et al., 2003; van Uhm, 2016), and there are prominent examples of species having gone extinct, at least in part, because of unsustainable use (e.g., the passenger pigeon *Ectopistes migratorius*; Hung et al., 2014). Species of plants, animals, and fungi may be harvested for sale to, or exchange with, others ('wildlife

trade') (Roe et al., 2002), or exploited for personal use by the harvester or their family with no transactions taking place ('subsistence use'). Commercial wildlife trade may take place legally or illegally in local, domestic, regional, or international markets, and may include diverse products traded for different purposes, including food, medicines, ornaments, fuel, or culture ('t Sas-Rolfes et al., 2019). Due to concerns about the impact of unsustainable extraction of species, various attempts have been made in the last century to regulate the use and trade of species at different scales. These range from measures to protect certain taxa at key sites (e.g., protected areas), the enactment of national laws restricting or prohibiting the harvest of species, to multilateral environmental agreements, including the Convention on International Trade in Endangered Species of Wild Fauna and Flora, which regulates international trade in about 39,000 species ('t Sas-Rolfes et al., 2019).

Assessing the links between wildlife trade, overexploitation, and extinction is complex. The first cause of complexity is that wildlife trade can have positive or negative effects on the conservation of traded species and ecosystems. Exploitation for the wildlife trade can threaten species but equally may contribute to conservation by providing economic incentives to conserve both species and their habitat. To achieve ecological sustainability, that is, use and/or trade does not denigrate biodiversity at the species or ecosystem level (Freese, 1997), requires an understanding of species populations, harvest rates, the impact of harvest (e.g., on density dependence), and the impact on wider ecosystems (e.g., trophic cascades) (Sutherland, 2001), among a broad range of social, economic, and governance factors (Cooney et al., 2015). Second, extinction is multidimensional, and it can be challenging to distinguish the impact of (over)exploitation from other threats. Species go extinct when the last individual has died, but there are other forms of extinction. Species may be commercially extinct, that is, it is no longer worthwhile harvesting them for profit. Species may be functionally extinct, that is, the species is no longer abundant enough to perform its ecosystem role or provide ecosystem services. Extinction theory indicates that driving species to extinction is not inherently simple, in part because the cost of finding the last individual of a species increases as the population declines (Courchamp et al., 2006). The societal extinction of species has also been recognised (Jarić et al., 2022).

Despite this complexity, wildlife trade is frequently characterised as a major threat to many species (e.g., Scheffers et al., 2019; Hughes, 2021), often without a supporting evidence base (Challender et al., 2022). The literature indicates that there are cases of both sustainable (e.g., southern white rhinos *Ceratotherium simum simum* in southern Africa; 't Sas-Rolfes et al., 2022) and unsustainable uses of wildlife (Marsh et al., 2022) and that assessing sustainability is difficult for many taxa because of a lack of data on life histories and populations, trade volumes, and the likely impacts of harvest (Smith et al., 2011). Social and economic factors related to the use and trade of species also require consideration (Cooney et al., 2015) because they can precipitate the overexploitation and extinction of species (e.g., the anthropogenic Allee effect; Courchamp et al., 2006; Lyons and Natusch, 2013), but understanding the conditions under which sustainability of use and trade in wildlife is, or can be, achieved is also inherently complex.

In this article, we examine the evidence base on species extinctions and links to wildlife trade by reviewing the published literature and the IUCN Red List of Threatened Species (hereafter 'Red List'). Due to time constraints, we conducted a rapid review, a streamlined version of a systematic review, that aims to synthesise

available evidence over a shorter period while maintaining key aspects of a systematic approach (Ganann et al., 2010). We searched the Scopus database for literature from all years between 1960 and 2021 inclusive that matched the search terms: (extinct* AND ['wildlife trade' OR 'wildlife traffick*' OR poach* OR over-harvest* OR overharvest* OR overhunt* OR over-hunt* OR overfish* OR over-fish* OR over-exploit* OR overexploit*]). We developed and refined our keywords based on multiple rounds of pilot searches. However, one trade-off of using rapid reviews is the potential for the omission of some articles (Ganann et al., 2010), and we note that we may have missed some examples of extinction where certain keywords were not used (e.g., where species were only described as 'lost' from an area, rather than extinct). For the Red List assessments, we downloaded data from the Red List version 2022-1 on all 200 taxa categorised as Extinct or Extinct in the Wild where Biological Resource Use (BRU) was coded as a threat. BRU refers to a number of threats on the Red List Threats Classification Scheme (IUCN, 2022), including hunting, collecting terrestrial plants, logging and wood harvesting, and fishing and harvesting aquatic resources, for which use may be intentional at different scales, the effects of use may be unintentional at different scales, and/or the motivations for use may be unknown.

Our literature search resulted in 1,769 documents, of which 1,698 were determined to be potentially relevant to our topic based on the title and abstract (see the Supplementary Material for all papers). A further 123 articles could not be accessed, although 13 of these were included based on information in the abstract alone. We read the abstract and, if necessary, the text of each paper to determine if the source mentioned: (1) links between trade and extinction (and if so which species, scale of extinction, scale of trade, and other contributing threats), (2) does the source mention the specific species driven to near-extinction from harvest for trade/trade (rather than subsistence use, and if so, which species), and (3) if any indirect threats from trade were mentioned in the source. For the Red List, many BRU threat codes referred only to subsistence use, so we used the assessment text, particularly the 'Rationale', 'Threats', and 'Use and trade' fields to determine whether commercial trade played a role in the species extinction rather than solely subsistence harvesting. Where only an ambiguous activity such as 'hunting' was noted as contributing to extinction, this was not coded as trade-related unless further details were reported on the commercial drivers of these activities.

## General narratives around wildlife trade and extinction

In total, 389 articles in our literature review mentioned general links between wildlife trade and extinction that did not provide details of specific extinction events. These included statements naming trade as the leading cause of existing extinctions, for example, '*Wildlife trade has become one of the main causes of species loss and extinction*' (Maulany et al., 2021), and '*indiscriminate poaching and illegal trade are becoming the main driver of species extinctions, more so than deforestation [in Asia]*' (Gomez et al., 2021). In addition, wildlife trade was named as a driver of extinctions that are still in progress, for example, '*Legal and illegal wildlife trade is a multibillion dollar industry that is driving several species toward extinction*' (Fukushima et al., 2020). Specific taxonomic groups were named as being at particular risk of trade-related extinction, including reptiles, where trade was said to have '*already driven at least 21 reptile species to extinction*' (Hughes et al., 2021). Finally, some statements linked specific markets and trade purposes to the extinction of certain taxa, for example, '*[lions, leopards and tigers] are*

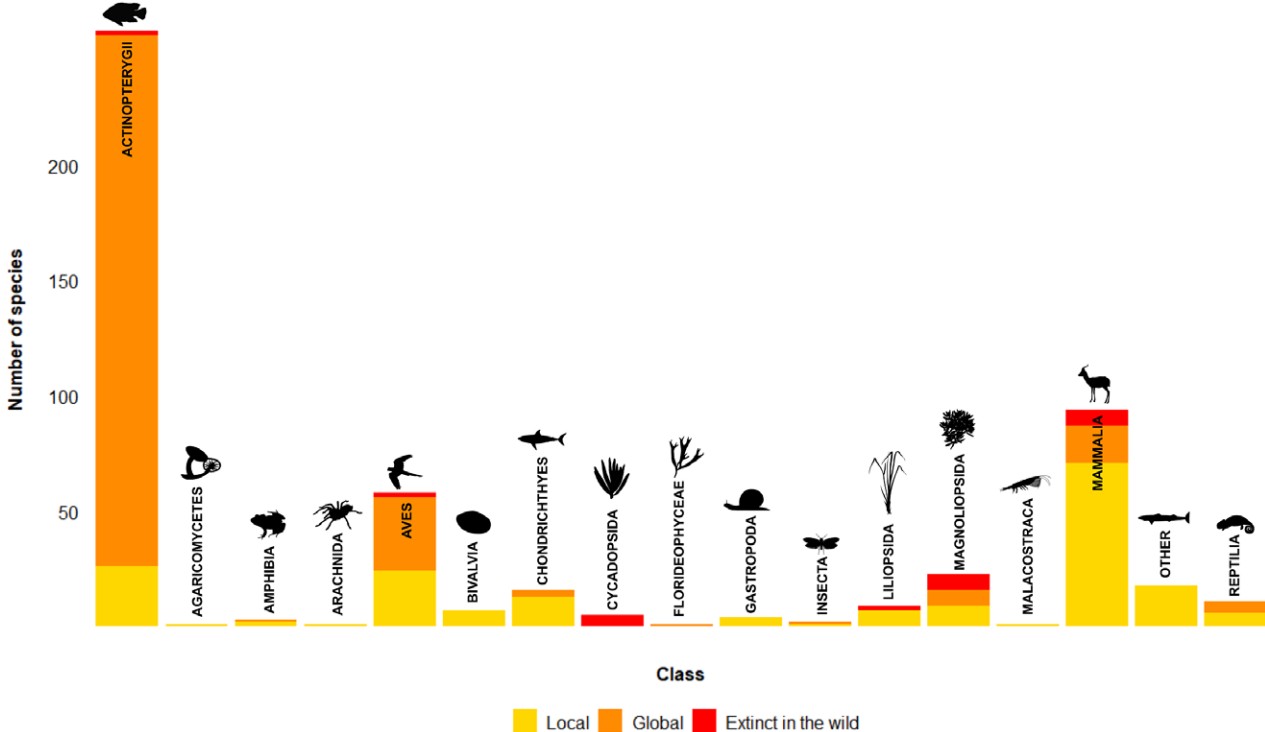

**Figure 1.** Species reported to have gone extinct due to trade or harvest for trade, by class. Based on the literature reviewed and species categorised as Extinct on the IUCN Red List of Threatened Species Version 2022-1.

*continuously facing the danger of extinction mainly due to poaching and hunting for their body parts, which are being greatly valued by apothecaries marketing traditional Chinese medicine*' (Singh et al., 2004). Only 110 articles included the scale of trade linked to extinction, including 21 reporting local trade as the only driver, 7 reporting only domestic trade, 56 reporting only international markets, and 27 reporting some combination of different market scales.

### Evidence of extinctions linked to wildlife trade

In total, 511 taxa were reported as extinct at some level due to trade-related factors with 294 reports of global extinctions, 25 species extinct in the wild, and 192 local, regional, or population extinctions (hereafter 'local extinctions'; see the Supplementary Material for all taxa). The Red List included 62 taxa assessed as Extinct and 16 as Extinct in the Wild, where trade-related threats were reported as a contributing factor. The literature search resulted in 224 articles that reported specific trade-related extinctions, relating to 451 unique taxa (243 global extinctions, 16 extinctions in the wild, and 192 local extinctions). A further 25 reports of extinction from the literature were excluded from final totals as the number of extinct taxa was not specified (e.g., *'due to exploitation, illegal fisheries, and trade…local extinctions have been recorded'*; Heinrich et al., 2019). In addition, 78 reported extinctions in the literature had too few details to assign to a specific taxonomic group and/or extinction scale (e.g., *'21 species* [of macrobiota in the Wadden sea] *were considered extinct in the twentieth century'*; Lotze et al., 2005). Similarly, one species of fish was omitted as it was reported only as commercially extinct. A further eight extinctions reported at the genus level were assumed to overlap with extinctions reported in other sources at the species level, for example, *'Within the African*

*cycads, four species of* Encephalartos *are already extinct in the wild'* (Ndou et al., 2021).

Only 18 specific reported extinctions were found both in the Red List and literature, with a further 60 reported only in the Red List, and 434 only in the literature. Some differences were due to differing reports of extinction drivers, including the huia *Heteralocha acutirostris,* which was reported in the literature to be extinct due to hunting for scientific specimens (Fernández-Palacios et al., 2021), but by the Red List as extinct due to habitat loss *'possibly along with hunting'* (Birdlife International, 2017), which we did not code as trade-related. While we included these cases in our final total if one source reported trade-related drivers, these differences highlight that exact causes of extinction are often unclear.

### Extinctions linked to the wildlife trade

Global/in the wild extinctions were reported for 297 animal taxa from seven classes, and 22 plant taxa from four classes (Figure 1). In addition, local extinctions were reported for 175 animal taxa from 11 classes, 16 plant taxa from two classes, and one fungal species.

Ray-finned fishes (class Actinopterygii) had the highest number of reported global/in the wild trade-related extinctions. Of the reported 230 extinctions, 200 were haplochromine cichlids in the African great lakes linked to the introduction of predatory species such as Nile perch (*Lates niloticus*) for the fishing industry (Witte et al., 1992; Ogutu-Ohwayo et al., 1997). While some cichlid species may have reappeared, and doubts about the exact role of the Nile perch in extinctions have been raised (van Zwieten et al., 2016), we include them here due to continuing claims that the perch was the main driver (e.g., Marshall, 2018). Similarly, 15 *Barbodes* species in the Philippines became extinct, partly due to introduced predatory fish and unsustainable fishing (e.g., Bitungu *Barbodes palaemophagus*; Torres et al.,

2020). The 26 local extinctions in this class include high-value sturgeon species targeted by overfishing, such as the Atlantic sturgeon *Acipenser sturio*, starry sturgeon *A. stellatus*, and beluga *Huso huso* (Bloesch et al., 2006). In addition, three cartilaginous fish taxa (class Chondrichthyes) were reported as 'likely to be extinct' due to overfishing: the Red Sea torpedo *Torpedo suessii*, the Java stingaree *Urolophus javanicus,* and the lost shark *Carcharhinus obsoletus* (Dulvy et al., 2021). Thirteen cartilaginous fish local extinctions were also reported, including the sawfish *Pristis pristis* and *P. pectinata* in Guinea-Bissau (Leeney and Poncelet, 2015), and the monkfish *Squatina squatina* in the Adriatic Sea (Holcer and Lazar, 2017).

Twenty-three mammal (class Mammalia) taxa and 34 bird (class Aves) taxa were reported as extinct globally/in the wild with links to trade. Several were linked to trade following European colonisation of islands in the 1800s, such as sea mink *Neovison macrodon* hunted for the fur trade (Helgen and Turvey, 2016), and the great auk *Pinguinus impennis* hunted for multiple products and for scientific specimens (Birdlife International, 2021). They also include recent high-profile megafaunal extinctions linked to international trade, such as the Northern white rhinoceros *Ceratotherium cottoni*, the Western subspecies of black rhinoceros *Diceros bicornis* subsp. *longipes,* and the Vietnamese subspecies of Javan rhinoceros *Rhinoceros sondaicus* subsp. *annamiticus* (Bennett, 2011). The 71 reported local extinctions of mammals linked to trade included small taxa such as the long-tailed chinchilla *Chinchilla lanigera* reportedly hunted to extinction in several areas of Chile for its fur (Jiménez, 1996), musk deer *Moschus* spp. hunted for traditional medicine trade in China (Zhou et al., 2004), and multiple reports of local extinctions of grey whales *Eschrichtius robustus* (Monte-Luna et al., 2007). The 24 reported local bird extinctions ranged from ostriches *Struthio* spp. in Syria due to hunting and commercial feather collecting (Brooke, 1995), to songbirds like the orange-headed thrush *Zoothera citrina* and white-rumped shama *Copsychus malabaricus*, collected for the live bird trade in Indonesia (Jepson and Ladle, 2009).

Four of the five reported global extinctions of reptiles (class Reptilia) were tortoises, including the Yunnan box turtle *Cuora yunnanensis* harvested for the illegal trade (Revenga et al., 2005), and the Burmese star tortoise *Geochelone platynota* reported as functionally extinct due to collection for wildlife markets (Raphael et al., 2019). In addition, overharvesting for scientific trade reportedly contributed to the extinction of the Cape Verde giant skink *Chioninia coctei* (Vasconcelos, 2022). The only globally extinct amphibian species (class Amphibia) reported was the pass stubfoot toad *Atelopus senex*, potentially linked to collection for the pet trade (IUCN SSC Amphibian Specialist Group, 2020). Six local reptile extinctions included the Nile crocodile *Crocodylus niloticus* (Bishop et al., 2009) and Nile monitor *Varanus niloticus* (Dowell et al., 2015), both reportedly due to overexploitation for skins. Two reported local amphibian extinctions were the Chinese giant salamander *Andras davidianus* harvested to stock farms as a luxury food (Turvey et al., 2018), and the lowland leopard frog *Lithobates yavapaiensis*, due to the introduction of the American bullfrog *L. catesbeianus* for food (Frías-Alvarez et al., 2010)

Fewer reports of trade-related plant extinctions were found. They included 14 globally extinct/extinct in the wild taxa in the dicots (class Magnoliopsida), such as the Chile sandalwood *Santalum fernandezianum*, due to logging for its aromatic timber (WCMC, 1998), and the extinction due, in part, to botanical collecting of pensée de cry *Viola cryana* in France in 1927 (Juillet, 2011), and *Lepidium obtusatum* in New Zealand (de Lange, 2014). Two species of monocots (class Liliopsida), Sprenger's tulip *Tulipa sprengeri* and the Chilean blue crocus *Tecophilaea cyanocrocus*, were reported extinct in the wild due

to harvest for the bulb trade (Maunder et al., 2001). Over-collection for horticultural trade also led to the extinction in the wild of five cycad species (class Cycadopsida) (*Encephalartos brevifoliolatus, E. heenanii, E. nubimontanus, E. relictus,* and *E. woodii)* between 1916 and 2006 (Bösenberg, 2022a,b,c,d,e). One species of red algae (class Florideophyceae), the Bennet's seaweed *Vanvoorstia bennettiana*, became extinct, in part, due to destruction and bycatch by commercial fisheries (Millar, 2003). Sixteen local plant extinctions (eight each of monocots and dicots) included three species of Asian slipper orchids (*Paphiopedilum canhii, P. vietnamese,* and *P. charlesworthii*) over-collected for the horticultural trade (Li et al., 2018), Brazilian rosewood *Aniba rosaeodora* cut for its timber (Sullivan and Swingland, 2006), and bitter kola *Garcinia kola* harvested for medicinal and edible use in Benin (Dadjo et al., 2020).

No global extinctions of invertebrates or fungi were reported, except for one species of insect (class Insecta), the Chilean stag beetle *Sclerostomulus nitidus*, reported to be extinct due to over-collection for the pet trade. However, while we include this as a reported extinction, recent reports suggest that it has been rediscovered (Crespin and Barahona-Segovia, 2021; Kehoe et al., 2021). More local extinctions were reported for invertebrates, including eight bivalve taxa (Class Bivalvia) of which four were species of giant clam *Hippopus hippopus*, *H. porcellanus*, *Tridacna gigas*, and *T. maxima* (Zann, 1994; Frias-Torres, 2017). Four species of gastropod (class Gastropoda) were reported as locally extinct, including the common whelk *Buccinum undatum*, which was extirpated in the Wadden sea due to the indirect effects of the fishing industry (Cadée et al., 1995) and the giant Mexican limpet *Scutellastra mexicana*, collected for its meat (Valdez-Cibrián et al., 2021). One species of insect, the mopani worm *Gonimbrasia belina*, experienced local extinction due to harvesting for food in Zambia (Ghaly, 2009), whereas one Arachnid, the Mexican red-knee tarantula *Brachypelma smithii,* was collected to local extinction for the pet trade in Acapulco (Fukushima et al., 2019). One species of crustacean (class Malacostraca), the white-clawed crayfish *Austropotamobius pallipes*, went locally extinct in several areas of Europe due to overfishing or introduced commercial species (e.g., Italy; Endrizzi et al., 2013).

The only reported extinction in the fungal Kingdom was the local extinction in the Alps of the agarikon *Fomitopsis officinalis* (*Laricifomes officinalis* in the source, class Agaricomycetes), over-harvested in the early twentieth century due to demand from the pharmaceutical industry for its antiviral and antibacterial properties (Girometta et al., 2021).

## Near-extinctions

In addition to local or global extinction, 340 taxa in 19 classes were reported to be nearing extinction at some scale due to trade-related threats (Figure 2). Mammals had the highest number of reports of near-extinction (95), with high numbers also reported in ray-finned fishes (68), dicots (43), and birds (38). While many reported near-extinctions were high-profile mammal species, near-extinctions spanned all taxonomic kingdoms, and included taxa in classes with no reported global or local extinctions. These include the invertebrate classes of sea cucumbers (class Holothuroidea), anthozoans (class Anthozoa), and arachnids (Class Arachnida), as well as plants in the conifers (class Pinopsida) and club mosses (class Lycopodiopsida), and the fungal class Sordariomycetes. However, beyond the number of examples of near-extinction reported in the literature, it is difficult to judge the scale or imminence of the threat of future extinctions from trade-related drivers, as near-extinctions were often reported in vague or subjective terms. Phrases such as

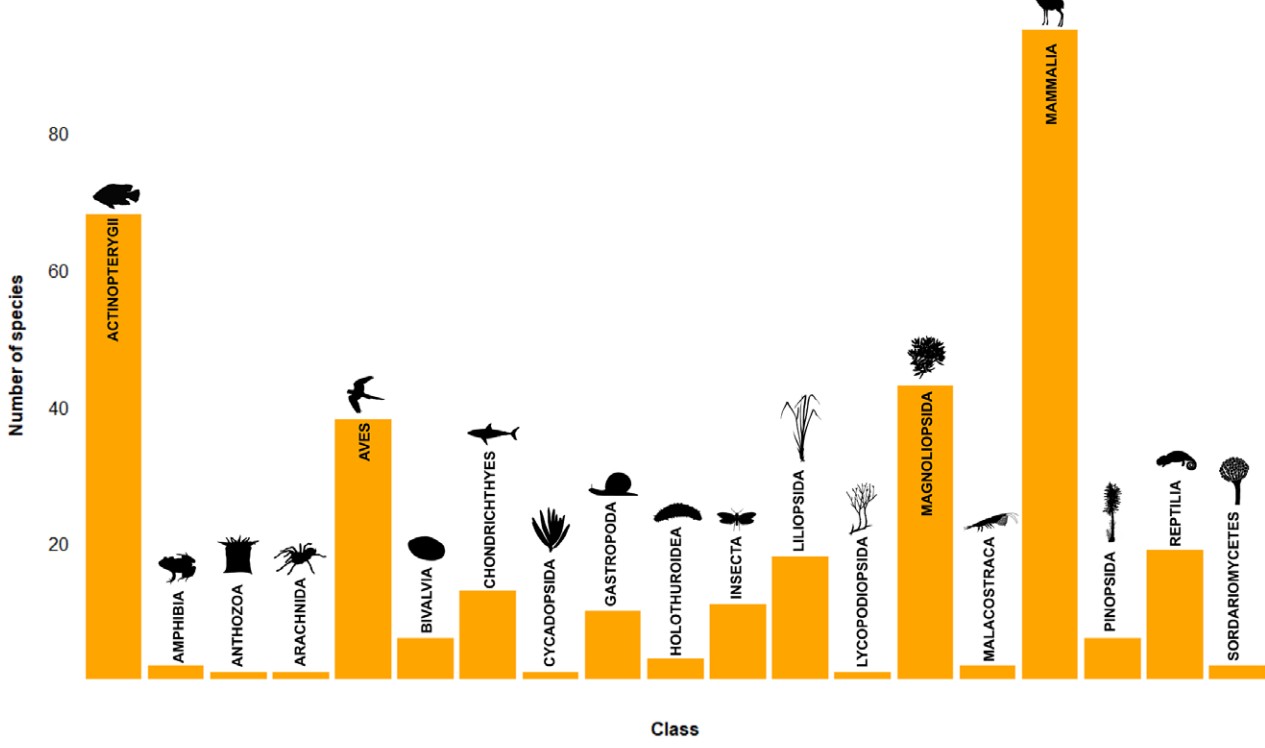

**Figure 2.** Species reported to be 'near extinct' or similar due to trade or harvest for trade, by class. Based on the literature review conducted.

'on the brink of extinction' or 'being driven to extinction' were common, along with more emotive phrasing, for example, '*the Chinese pangolin is being conveyed along this lonely corridor towards extinction because some consumers attach medicinal, nutritional and cultural value to their body parts*' (Aisher, 2016).

### Trade-related drivers of extinction can be direct or indirect

Many high-profile extinctions and near-extinctions related to the wildlife trade are linked to the direct effects of harvesting, which, if unsustainable, directly removes individuals from a population at a rate higher than the species can sustain. However, we found multiple examples of indirect threats linked to the wildlife trade that have led to extinction or near-extinction of taxa, including those that are not traded themselves. Notable examples relate to the bycatch of non-target taxa, including marine mammals such as the vaquita *Phocoena sinus*, which is near-extinction due to illegal totoaba *Totoaba macdonaldi* fisheries in Mexico (Jaramillo-Legorreta et al., 2017). While most bycatch examples are related to fisheries, the tooth-billed pigeon *Didunculus strigirostris* was also noted to be near-extinction due to bycatch in the Pacific pigeon *Ducula pacifica* wild meat trade (Stirnemann et al., 2018). As well as bycatch, non-target taxa may be destroyed during harvest, such as in the cases of the common whelk and Bennett's seaweed where extinction was linked to destruction by fishing gear and pollution from the fishing industry (Cadée et al., 1995; Millar, 2003). In some cases, subsistence use that would not exist without wildlife trade may lead to extinctions, such as the examples of the white swamphen *Porphyrio albus* (Birdlife International, 2016a), the Floreana giant tortoise *Chelonoidis niger* (van Dijk et al., 2017), and the Amsterdam duck *Anas marecula* (BirdLife International., 2016b), which were all hunted to extinction to provide provisions for commercial whalers or sealers.

In addition to indirect impacts of harvest, the movement of individuals of traded taxa can threaten other species that rely on them in their native habitats, or that are threatened by their introduction into new areas. For example, the large-scale harvest of Atlantic horseshoe crabs *Limulus polyphemus* for commercial fishing bait is removing the key food source of the red knot *Calidris canutus rufa* in Delaware Bay, USA, which is threatening the subspecies with extinction (Baker et al., 2004). Similarly, the over-harvest of seed-dispersers for trade can threaten tree species that rely on them, such as in the case of the large canopy tree *Choerospondias axillaris*, which is likely to go extinct in some areas where mammalian seed-dispersers have been removed for the wild meat trade (Brodie et al., 2009). In addition to the removal of species, the introduction of a new species can also lead to extinctions. The highest number of extinctions found during our review related to the introduction of non-native species for trade, such as the Nile perch that reportedly led to the extinction of 200 species of cichlids, and the local extinction of the white-clawed crayfish linked to the competition from the American crayfish *Orconectes limosus* introduced by the fishing industry (Endrizzi et al., 2013). The latter example also demonstrates another indirect threat – that of diseases or parasites carried by introduced commercial species, as the American crayfish carried a parasite that contributed to local extinctions of the native crayfish species (Endrizzi et al., 2013). Similarly, while few clear cases of amphibian extinctions linked to trade were reported, concerns have been raised about the global amphibian trade's role in spreading chytrid fungus and other diseases (Kolby, 2018; Hughes et al., 2021).

### The uncertain role of trade-related threats in extinctions

A major challenge in defining the number of extinctions linked to trade is the lack of data available on causes of extinction in the

literature, especially in older examples. In addition, potential taxonomic biases in reporting are likely to have led to higher numbers of animal taxa reported as extinct, or having the reasons for their extinctions fully examined, than plants or fungi. While a species may be reported as likely threatened due to trade, the harvest of a species for trade does not automatically present an extinction risk, and other threats such as habitat loss may be sufficient to cause the extinction of traded species (Moyle, 1998). Some taxa had trade-related threats listed as the only or primary cause of extinction, such as the Japanese sea lion *Zalophus japonicus,* the extinction of which was reportedly due to its capture for the circus trade and hunting for its skin, organs, oil, and whiskers (Lowry, 2017). In other cases, multiple trade-related threats combined, including the Caribbean monk seal *Neomonachus tropicalis,* where over-hunting for trade in oil and skins combined with persecution by the fishing industry led to its extinction (Lowry, 2015). Wildlife trade is complex, with diverse markets operating at many different scales, but the majority of articles did not include any details of the scale of trade that drove harvesting or other trade-related threats.

Often the exact cause of extinction is unknown, and the relative roles of trade compared to other threats could not be determined. For example, the Franklin tree *Franklinia alatamaha* was over-collected for the horticultural trade, but also is likely to have declined due to habitat loss (Rivers, 2015). Trade-related threats often act in combination with threats such as habitat loss, climate change, and overexploitation for subsistence use to cause extinction. For example, local extinctions of Atlantic sturgeon in the Danube were reportedly due to intensive commercial fishing combined with habitat loss and pollution (Bloesch et al., 2006), while multiple reports of tiger *Panthera tigris* extinctions linked to poaching for the illegal trade also noted the significant role of habitat loss and decline in prey species (e.g., Gopal et al., 2010; Lynam, 2010). In some cases where little data are available on wild populations or drivers of extinction, the exact role of trade in an extinction may have been disputed (e.g., Lake Victoria cichlids; van Zwieten et al., 2016) or extinct species reportedly rediscovered (e.g., Chilean stag beetle *S. nitidus*; Crespin and Barahona-Segovia, 2021; Kehoe et al., 2021), leading to some uncertainty. In addition, differentiating trade from subsistence use is challenging if enough detail is not provided, with many sources naming only hunting, fishing, or overexploitation as a threat, rather than the driver behind these activities. In our review, we tried to use context and other literature to verify whether extinctions were due to trade, but our totals are still likely to have omitted some extinctions where drivers were unclear, or included trade-related extinctions that were later disputed. As such, our review should not be considered exhaustive, and the number of extinctions we report should be considered an estimate.

## Conclusions and recommendations

We show that there is evidence that wildlife trade is linked to the extinction of multiple wild taxa, and is contributing to the extinction risk of many more. However, clear links naming trade-related threats (e.g., harvesting for trade in particular as distinct from subsistence use) as a primary driver of extinction are rare, and their contribution to the decline of species is frequently difficult to determine. Furthermore, reporting bias is likely to mean that trade-related extinctions are underestimated for lesser-studied but highly traded taxonomic groups, such as fungi, plants, and invertebrates. Exaggerated reports of extinctions or extinction risk

that do not match the evidence may attract funding and attention for a species, but may distract from other threats (Koot, 2021). In addition, care should be taken when reporting extinction and near-extinction due to the unintended consequences this can have. For example, 'extinct' species are sometimes rediscovered, which can lead to increased harvesting in some markets due to demand for rarity (Crespin and Barahona-Segovia, 2021). Better data and more accurate characterisation of the links between harvest, trade (at various levels), and the extinction risk for species are much needed to better understand the impact of wildlife trade on species populations and policies designed to ensure that the harvest, use, and trade of wildlife in the future are sustainable. This is also important to avoid mischaracterising wildlife trade and the impact it may have on species populations, and misleading policy processes (Natusch et al., 2021; Challender et al., 2022).

In this context, we make the following recommendations:

1. Research on wildlife trade and links to extinction should pay particular attention to accurately characterising the species that are harvested for trade, and the scale of that trade (including local, domestic, and international levels), as well as the impact of harvest on species populations; and the relative role that this harvest for trade may play in extinction risk. Uncertainty should be explicitly recognised to allow gaps in evidence to be clearly identified.

2. Where such evidence is available, more objective assessments of extinction risk linked to trade should be prioritised. This may involve more in-depth assessments of how various trade-related risk factors may affect specific taxa (e.g., McClenachan et al., 2016), modelling to assess extinction risk under different trade scenarios (e.g., Creel et al., 2016), or novel methods from other disciplines to assess risk from different combinations of factors (e.g., Zheng et al., 2022). Objective assessments could subsequently inform policy decisions and contribute to discussions about sustainability of trade in different taxa and systems.

3. Research should also seek to characterise wildlife trade to distinguish between different forms of trade (e.g., different product types and markets), the direct and indirect impact of different trade types on targeted species and wider biodiversity, and the effectiveness of different interventions (including both regulatory and non-regulatory measures), to better understand the determinants of sustainability, including whether particular types of trade are more harmful than others.

4. Finally, researchers should pay careful attention to the language used in reports of extinction or extinction risk linked to trade, to ensure that any uncertainty is recognised (Challender et al., 2022). In addition, hyperbolic and imprecise statements about the impact of harvest and/or trade on particular species or extinction risk more broadly should be avoided, including reports of extinctions for which there is insufficient evidence. Doing so could avoid diverting attention from species that are more at risk from trade but lesser known (e.g., plants) or from other threats that may be more pressing than those related to trade.

**Open peer review.** To view the open peer review materials for this article, please visit http://doi.org/10.1017/ext.2023.7.

**Supplementary Materials.** The supplementary material for this article can be found at http://doi.org/10.1017/ext.2023.7.

**Data availability statement.** The authors confirm that the data supporting the findings of this study are available within the Supplementary Material.

**Acknowledgements.** The authors acknowledge Griensteidl and T. Michael Keesey for use of the Lycopodiopsida silhouette in Figure 2 under this licence (https://creativecommons.org/licenses/by-sa/3.0/) without changes.

**Author contribution.** A.H. and D.W.S.C. designed the work, interpreted the reviewed data and wrote the first draft. A.H. and J.W. reviewed the Red List data. All authors reviewed the literature and contributed to the final text.

**Financial support.** This research received no specific grant from any funding agency, commercial or not-for-profit sectors. However, authors' time was supported by the Kadas Senior Research Fellowship at Worcester College, Oxford (to A.H., with support for the time of A.R.D., R.O. and J.W.). D.W.S.C. acknowledges funding from the UK Research and Innovation's Global Challenges Research Fund (UKRI GCRF) through the Trade, Development and the Environment Hub project (project number ES/S008160/1). T.K. was supported by the Overseas Research Fellowships by the Japan Society for the Promotion of Science (JSPS).

**Competing interest.** The authors declare that they have no competing interests.

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
