## [Reviewer Report]

*Comments to Author*: General comments:

I enjoyed reading your article titled ‘Trading species to extinction: evidence of extinction linked to the wildlife trade’. The article is well-written and makes an important contribution on the evidence base on species extinctions and links to wildlife trade. As you will see I have only few specific comments below, which are important to include in the interest of acknowledging some of the limitations of the study and explaining the rationale behind the study better.

Specific comments:

line 92: what type of literature review? Systematic? Or? If not systematic, please do acknowledge some of the limitations of not using this approach as part of the Methods or Discussion sections.

line 101: most of the readers of ‘Extinction’ will not be familiar with the term ‘biological resource use’. Please explain in lay words what this threat is and what threat categories it includes (e.g., hunting, trapping, etc.). In addition, for species classified as Extinct or Extinct in the Wild in the IUCN Red List of Threatened Species, did you consider their extinctions to be linked, at least in part, to wildlife trade in case the threat was ‘Hunting & collecting terrestrial animals’? Can ‘Hunting & collecting terrestrial animals’ be considered wildlife trade?

line 105-106: need to mention that the list of papers is provided as an Appendix file? Also, could you provide, as part of this file, the abstract and DOI for each article?

---

## [Reviewer Report]

*Comments to Author*: Trading species to extinction: evidence of extinction linked to the wildlife trade

Reviewer Report

This is an interesting and potentially important paper. The authors should be congratulated on making a rigorous and needed search into extant reported cases of different forms of extinction and their association with trade.

Before this paper is published I would encourage them to be more precise about their key term, as this would then allow the empirical insights also to be matched by conceptual rigour.

To put this briefly: what do they mean by ‘trade’? The authors are careful to distinguish different types of extinction (line 68) – can they demonstrate similar sophistication about the different forms and variety of trade? This is mentioned without qualification (lines 53, 59, 61). But there are many types of trade, as indeed the authors allude to. For instance they mention a distinction between subsistence us and trade on lines 340-342. But even ‘subsistence’ is varied and is rarely totally autonomous. It often entails all sorts of exchange between families. That is why hunting can be a source of prestige within communities – the meat is shared. Likewise (and more generally where the trade involves money) it can incorporate different spatial scales – local, regional, national, international; different variations of intensity of harvesting with respect to the availability of the wildlife; different dynamics seasonally, and over longer time periods; different forms of regulation (legal but none, illegal and none; illegal and governed by criminal gangs, regulated by legislation, subject to certification processes etc etc); different purposes (food, clothing, pleasure/leisure) and so on.

My suggestion is that the authors explain the complexity and varieties of trade and then, if possible, consider how well the forms of trade that have had a role in extinction have been described. My suspicion is that the literature does not describe trade well. This is one of the gaps. But it will be important to show that. The authors are surely right that more work needs to be done on the role of trade in extinction, but accurately characterising the form of trade will be essential for achieving the objective assessments that they desire.

The corollary to this, and my second encouragement, is that I think that the authors could be much more ambitious with respect to the research agenda that they set. If species are being traded but only becoming locally extinct – does that mean that they are being traded elsewhere and not becoming extinct? Or are there forms of trade which are sustained for a long time, but then become unsustainable due to social / ecological changes. By distinguishing between different forms of trade and their consequences for wildlife over time this could allow us to understand better what forms of trade, and what forms of regulation, produce beneficial results and which prove disastrous. That would be an interesting research agenda and it would be great if the authors could suggest it.

Dan Brockington

Barcelona, November 2022

Minor points

Line 95 – were you missing a ‘*’ after traffick in your search term “wildlife trade” OR “wildlife traffick”? Does that miss “wildlife trafficking”?

Lines 170-172; 305-6. Please double-check how extinct the Lake Victorian cichlids actually are, and the cause of their (temporary?) disappearance. More recent research suggests that the cichlids had been predating young perch populations until an El Nino event and increasing phosphorous in the water changed the diatom composition. See van Zwieten et al Can. J. Fish. Aquat. Sci. 73: 1–22 (2016) dx.doi.org/10.1139/cjfas-2015-0130

---

## [Editor Report]

*Comments to Author*: I agree with the reviewers that the manuscript is well developed and with a good potential to become an important contribution to the field, but I also agree that there is room for improvement, especially to consider better different types of trade and their relationship with the extinction process. 

Reviewer 1:

This is an interesting and potentially important paper. The authors should be congratulated on making a rigorous and needed search into extant reported cases of different forms of extinction and their association with trade.

Before this paper is published I would encourage them to be more precise about their key term, as this would then allow the empirical insights also to be matched by conceptual rigour. 

To put this briefly: what do they mean by ‘trade’? The authors are careful to distinguish different types of extinction (line 68) – can they demonstrate similar sophistication about the different forms and variety of trade? This is mentioned without qualification (lines 53, 59, 61). But there are many types of trade, as indeed the authors allude to. For instance they mention a distinction between subsistence us and trade on lines 340-342. But even ‘subsistence’ is varied and is rarely totally autonomous. It often entails all sorts of exchange between families. That is why hunting can be a source of prestige within communities – the meat is shared. Likewise (and more generally where the trade involves money) it can incorporate different spatial scales – local, regional, national, international; different variations of intensity of harvesting with respect to the availability of the wildlife; different dynamics seasonally, and over longer time periods; different forms of regulation (legal but none, illegal and none; illegal and governed by criminal gangs, regulated by legislation, subject to certification processes etc etc); different purposes (food, clothing, pleasure/leisure) and so on.

My suggestion is that the authors explain the complexity and varieties of trade and then, if possible, consider how well the forms of trade that have had a role in extinction have been described. My suspicion is that the literature does not describe trade well. This is one of the gaps. But it will be important to show that. The authors are surely right that more work needs to be done on the role of trade in extinction, but accurately characterising the form of trade will be essential for achieving the objective assessments that they desire.

The corollary to this, and my second encouragement, is that I think that the authors could be much more ambitious with respect to the research agenda that they set. If species are being traded but only becoming locally extinct – does that mean that they are being traded elsewhere and not becoming extinct? Or are there forms of trade which are sustained for a long time, but then become unsustainable due to social / ecological changes. By distinguishing between different forms of trade and their consequences for wildlife over time this could allow us to understand better what forms of trade, and what forms of regulation, produce beneficial results and which prove disastrous. That would be an interesting research agenda and it would be great if the authors could suggest it.

Dan Brockington

Barcelona, November 2022

Minor points

Line 95 – were you missing a ‘*’ after traffick in your search term “wildlife trade” OR “wildlife traffick”? Does that miss “wildlife trafficking”?

Lines 170-172; 305-6. Please double-check how extinct the Lake Victorian cichlids actually are, and the cause of their (temporary?) disappearance. More recent research suggests that the cichlids had been predating young perch populations until an El Nino event and increasing phosphorous in the water changed the diatom composition. See van Zwieten et al Can. J. Fish. Aquat. Sci. 73: 1–22 (2016) dx.doi.org/10.1139/cjfas-2015-0130 

Reviewer 2:

General comments:

I enjoyed reading your article titled ‘Trading species to extinction: evidence of extinction linked to the wildlife trade’. The article is well-written and makes an important contribution on the evidence base on species extinctions and links to wildlife trade. As you will see I have only few specific comments below, which are important to include in the interest of acknowledging some of the limitations of the study and explaining the rationale behind the study better. 

Specific comments:

line 92: what type of literature review? Systematic? Or? If not systematic, please do acknowledge some of the limitations of not using this approach as part of the Methods or Discussion sections.

line 101: most of the readers of ‘Extinction’ will not be familiar with the term ‘biological resource use’. Please explain in lay words what this threat is and what threat categories it includes (e.g., hunting, trapping, etc.). In addition, for species classified as Extinct or Extinct in the Wild in the IUCN Red List of Threatened Species, did you consider their extinctions to be linked, at least in part, to wildlife trade in case the threat was ‘Hunting & collecting terrestrial animals’? Can ‘Hunting & collecting terrestrial animals’ be considered wildlife trade? 

line 105-106: need to mention that the list of papers is provided as an Appendix file? Also, could you provide, as part of this file, the abstract and DOI for each article?

---

## [Reviewer Report]

*Comments to Author*: Thank you for fully addressing my concerns. I believe the manuscript has improved and is now suitable for publication.

---

## [Editor Report]

*Comments to Author*: Thank you for submitting the revised version of your manuscript. I apologize for the delay with the review process, but we had problems with getting comments from one of the reviewers who reviewed the previous version of the manuscript. Based on the evaluation of the changes made in the manuscript and comments of reviewers, I think that the manuscript can be accepted in its present form. I noticed a number of typos in the text, but I expect that they will be dealt with during the pre-proof editing and the proof stage.